# ENPP1 variants in patients with GACI and PXE expand the clinical and genetic heterogeneity of heritable disorders of ectopic calcification

**Douglas Ralph**[1,2,3], **Yvonne Nitschke**[4], **Michael A. Levine**[5], **Matthew Caffet**[6], **Tamara Wurst**[6], **Amir Hossein Saeidian**[1,2], **Leila Youssefian**[1], **Hassan Vahidnezhad**[1], **Sharon F. Terry**[6], **Frank Rutsch**[4], **Jouni Uitto**[1,3], **Qiaoli Li**[1,3]*

1 Department of Dermatology and Cutaneous Biology, Sidney Kimmel Medical College, and Jefferson Institute of Molecular Medicine, Thomas Jefferson University, Philadelphia, Pennsylvania, United States of America, 2 Genetics, Genomics and Cancer Biology Ph.D. Program, Jefferson College of Life Sciences, Thomas Jefferson University, Philadelphia, Pennsylvania, United States of America, 3 PXE International Center of Excellence in Research and Clinical Care, Thomas Jefferson University, Philadelphia, Pennsylvania, United States of America, 4 Münster University Children's Hospital, Münster, Germany, 5 Division of Endocrinology and Diabetes, Children's Hospital of Philadelphia, Philadelphia, Pennsylvania, United States of America, 6 PXE International, Inc., Damascus, Maryland, United States of America

* Qiaoli.Li@Jefferson.edu

**Data Availability Statement:** All relevant data are within the paper and its Supporting Information files.

## Abstract

Pseudoxanthoma elasticum (PXE) and generalized arterial calcification of infancy (GACI) are clinically distinct genetic entities of ectopic calcification associated with differentially reduced circulating levels of inorganic pyrophosphate (PPi), a potent endogenous inhibitor of calcification. Variants in ENPP1, the gene mutated in GACI, have not been associated with classic PXE. Here we report the clinical, laboratory, and molecular evaluations of ten GACI and two PXE patients from five and two unrelated families registered in GACI Global and PXE International databases, respectively. All patients were found to carry biallelic variants in ENPP1. Among ten ENPP1 variants, one homozygous variant demonstrated uniparental disomy inheritance. Functional assessment of five previously unreported ENPP1 variants suggested pathogenicity. The two PXE patients, currently 57 and 27 years of age, had diagnostic features of PXE and had not manifested the GACI phenotype. The similarly reduced PPi plasma concentrations in the PXE and GACI patients in our study correlate poorly with their disease severity. This study demonstrates that in addition to GACI, ENPP1 variants can cause classic PXE, expanding the clinical and genetic heterogeneity of heritable ectopic calcification disorders. Furthermore, the results challenge the current prevailing concept that plasma PPi is the only factor governing the severity of ectopic calcification.

## Author summary

Biallelic inactivating mutations in the ENPP1 gene cause generalized arterial calcification of infancy (GACI), a frequently fatal disease characterized by infantile onset of widespread arterial calcification and/or narrowing of large and medium-sized vessels often resulting

**Funding:** This study was supported by PXE International and the National Institutes of Health/National Institute of Arthritis and Musculoskeletal and Skin Diseases grants R01AR072695 (to JU and QL) and R21AR077332 (to QL). The funders had no role in study design, data collection and analysis, decision to publish, or preparation of the manuscript.

**Competing interests:** The authors have declared that no competing interests exist.

in the early demise of affected individuals. Significantly reduced, almost zero plasma levels of a potent and endogenous calcification inhibitor, inorganic pyrophosphate (PPi), is thought to be the underlying cause of vascular calcification in GACI. Mutations in *ENPP1* have not been found in patients with pseudoxanthoma elasticum (PXE), another genetic multisystem ectopic calcification disorder caused by mutations in the *ABCC6* gene. This study reports that *ENPP1* mutations can also cause PXE with more favorable clinical outcomes. In addition, it was previously thought that plasma PPi levels correlate with vascular calcification severity. However, we here show that vascular calcification severity does not correlate with plasma PPi levels. The results suggest that in addition to PPi, the long-believed determinant of ectopic calcification, additional mechanisms may be at play in regulating ectopic calcification.

## Introduction

*ABCC6* and *ENPP1* encode proteins that are required for the generation of inorganic pyrophosphate (PPi), a potent endogenous inhibitor of calcification [1], and pathogenic variants in both genes have been associated with syndromes of ectopic calcification [2]. Biallelic inactivating variants in *ENPP1* or *ABCC6* cause generalized arterial calcification of infancy type 1 (OMIM 208000) and type 2 (OMIM 614473), respectively, rare autosomal recessive disorders that are nearly indistinguishable and often diagnosed by prenatal vascular calcification [3–5]. Arterial calcification and intimal hyperproliferation frequently lead to stenoses and early demise of affected infants by six months of age [6,7]. Loss-of-function variants in *ABCC6* also cause pseudoxanthoma elasticum (PXE; OMIM 264800), an autosomal recessive disorder characterized by late-onset yet progressive ectopic calcification in the skin, eyes, and arterial blood vessels [8]. In contrast to GACI, the clinical manifestations of PXE are usually not recognized until early adulthood or at adolescence, either diagnosed by practicing dermatologists finding yellowish papules of the skin that progressively coalesce to make a leathery plaque on flexor areas, or by the patient presenting with a retinal bleed. The skin manifestations usually signify later development of vascular complications [8].

The *ENPP1* gene encodes a type II transmembrane glycoprotein, the principal enzyme that generates extracellular PPi by hydrolysis of adenosine triphosphate (ATP). Reduced plasma PPi concentration, at approximately 0–10% of control subjects, is the basis for vascular calcification in ENPP1-deficiency [9]. ABCC6, a hepatic plasma membrane transporter, works upstream of ENPP1 by facilitating the extracellular release of ATP, the substrate of ENPP1, thus contributing to PPi generation as well [10,11]. As a result, plasma PPi levels in patients with PXE and *Abcc6* knockout murine models of PXE are reduced to approximately 30–50% of controls [10,12–14]. While GACI and PXE are considered PPi deficiency disorders, the plasma PPi concentrations, reduced to different extents, were thought to correlate with the onset and disease severity in these conditions [15,16].

Natural history studies of patients with GACI due to ENPP1-deficiency indicate that many who survive the critical first year of life experience some resolution of arterial calcification but also can later develop autosomal recessive hypophosphatemic rickets type 2 (ARHR2; OMIM 613312) [5]. In addition, some GACI patients with ENPP1-deficiency, diagnosed prenatally or neonatally with vascular calcification, have been reported to develop skin lesions and/or angioid streaks, features that occur in PXE [3,5,17]. These manifestations, however, appear later in adult life. Despite the considerable genotypic and phenotypic overlap between PXE and GACI, *ENPP1* variants have not been associated with classic PXE. Here we report the results of clinical, laboratory, and molecular evaluations of ten patients with GACI1 in five

distinct families and two patients with PXE in two unrelated families, all carrying biallelic variants in *ENPP1*. The results show that in addition to GACI, *ENPP1* variants can also cause PXE, expanding the phenotypic and genotypic overlap between GACI and PXE.

## Results

### Clinical features and biochemical findings of GACI patients

A total of 10 affected patients from five distinct families (families #1–5) with clinical manifestations consistent with GACI with or without ARHR2 were examined. These individuals are members of GACI Global. The nuclear pedigrees of these families are shown in Fig 1, and their clinical characteristics are shown in Fig 2 (panels a to e), and detailed in the S1 Text. These patients were diagnosed with GACI prenatally or neonatally due to extensive arterial calcification that was frequently associated with arterial stenosis, with or without ARHR2 at the time of examination. Reduced serum phosphorus levels, elevated alkaline phosphatase, and elevated or high-normal FGF23 were found in patients with ARHR2 (S1 Table), a finding frequently encountered in GACI patients beyond infancy [3,6,18]. Five out of 10 patients died of cardiac complications within four years of life, consistent with the often fatal outcome of these patients documented previously [3,6,18]. None of the 10 GACI patients had evidence of PXE-like skin lesions or retinopathy, *i.e.*, peau d'orange, angioid streaks, or optic disc drusen.

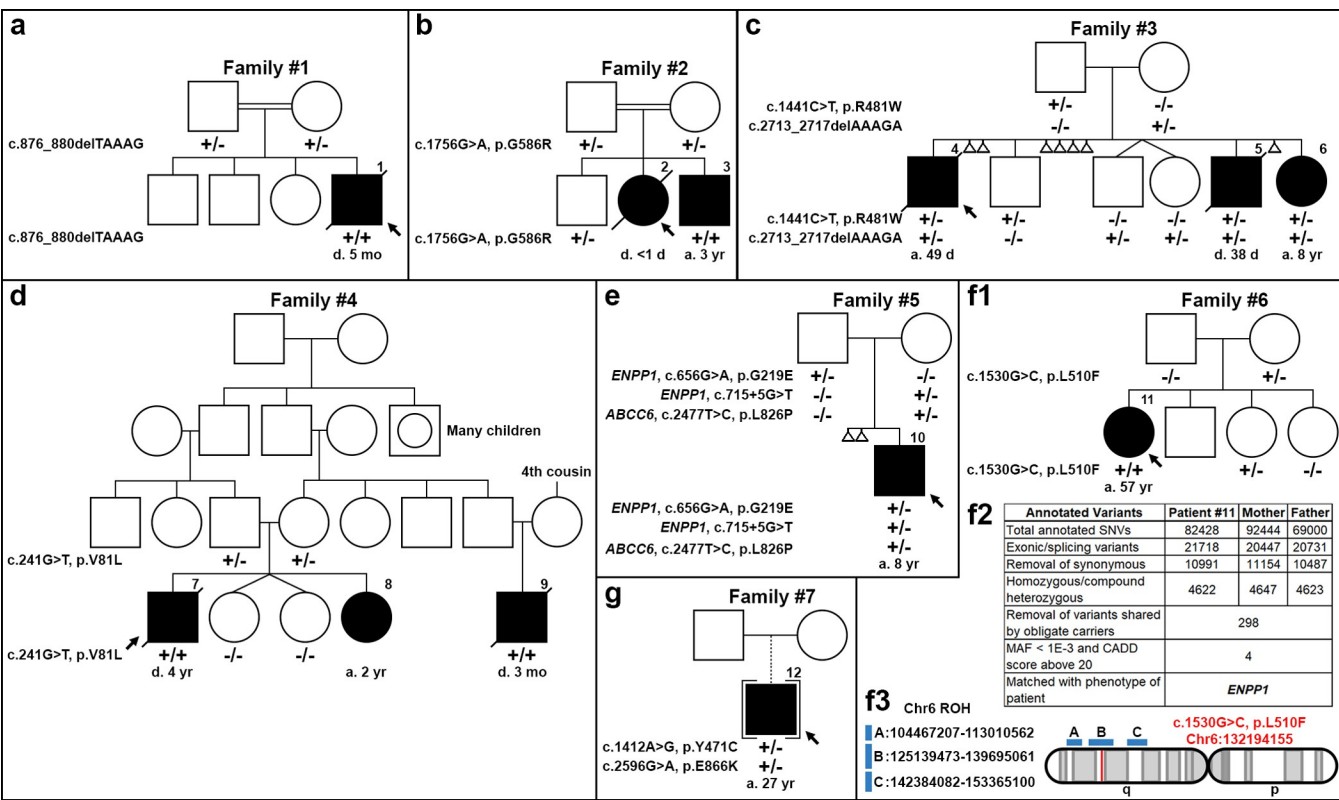

**Fig 1. Nuclear pedigrees of families with a diagnosis of GACI or PXE.** (a-e), GACI families #1–5; (f-g), PXE families #6–7. Patient identifier (#1 through 12) was placed above their symbol. The variants identified in the *ENPP1* and *ABCC6* genes in the individual family members are indicated below each individual: +/+, variants present in both alleles; +/-, heterozygous; -/-, homozygous for the wild-type allele. Unless otherwise noted, all variants were found in *ENPP1*. The stepwise bioinformatic filtering of exome sequencing data for variant detection in family #6 was narrowed to *ENPP1* (panel f2). The uniparental inheritance of the c.1530G>C (p.L510F) variant in *ENPP1* in patient #11 was shown in one Run of Homozygosity (ROH; blue region) on Chromosome 6 inherited from her mother (panel f3). d, died; a, alive.

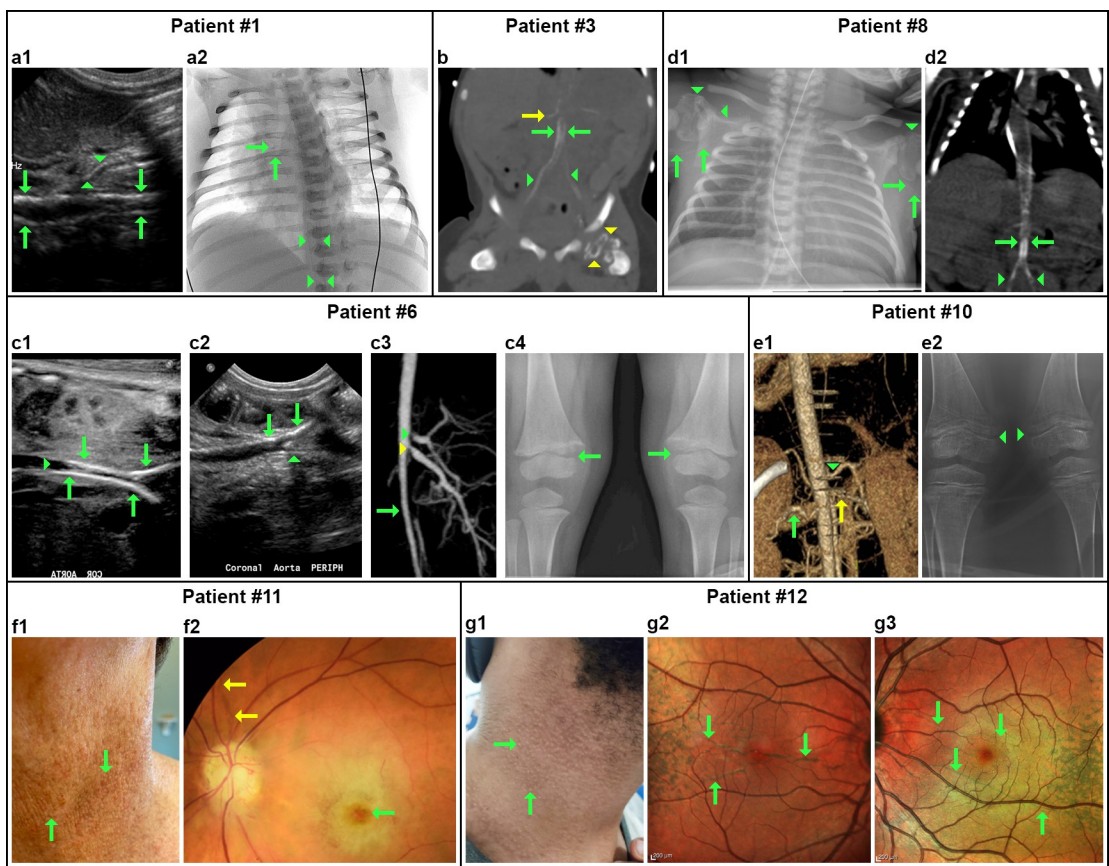

**Fig 2. Clinical features in individuals with GACI and/or hypophosphatemic rickets, or PXE.** Patient #1: (**a1**) Prenatal ultrasound of the upper abdomen at 35 weeks of gestation showed calcification along the wall of the aorta (green arrows) and proximal superior mesenteric artery (green arrowheads); (**a2**) Chest X-ray (inverted) at three days after birth showing calcification in the right pulmonary artery (green arrows) and subtle circumferential calcification throughout the descending and abdominal aorta (green arrowheads). Patient #3: (**b**) Three days after birth CTA showed extensive calcification of the abdominal aorta (green arrows) extending into bilateral iliac arteries (green arrowheads) and right renal artery (yellow arrow). There is also abnormal calcification of the left proximal femoral epiphysis (yellow arrowhead). Patient #6: (**c1**) Fetal ultrasound at 28 weeks of gestation showed extensive circumferential calcification involving the mid to distal aorta extending to the common iliac arteries (green arrows) with luminal narrowing (green arrowhead); (**c2**) Coronal ultrasound at 14 days of age showed reduced calcification now limited to the aortic bifurcation (green arrows) and the common iliac arteries (green arrowhead); (**c3**) At four months of age CTA showed stenosis of the abdominal aorta (green arrow), celiac (yellow arrowhead) and superior mesenteric arteries (green arrowhead); (**c4**) X-ray of both knees at 24 months revealed symmetric widening of the medial distal femoral epiphysis (green arrows). Patient #8: (**d1**) X-ray showed extensive calcification of both axillary arteries (green arrows) and dystrophic scapular calcification of the shoulders (green arrowheads); (**d2**) Abdominal CTA showed calcification of abdominal aorta (green arrow) extending into bilateral iliac arteries (green arrowheads). Patient #10: (**e1**) CTA At 6.5 years of age showed left renal artery stenosis and dilation (green arrow), abdominal aorta stenosis (green arrowhead), and a beaded right renal artery yellow arrow); (**e2**) X-ray of both knees showed rough cupping of the posterior end of bilateral tibial shaft (green arrowheads). Patient #11: (**f1**) Pseudoxanthomatous skin lesions on the neck (green arrows) at 57 years of age; (**f2**) The presence of angioid streaks (green arrow) and cherry red spot (yellow arrow) in the left eye of patient #11. Patient #12: (**g1**) Pseudoxanthomatous skin lesions on the neck (green arrows) at 27 years of age; (**g2, g3**) SPECTRALIS Multicolor Tacking Laser Tomography at 23 years of age demonstrated angioid streaks in both eyes (green arrows).

## Clinical features and biochemical findings of PXE patients

Previously, it has not been demonstrated that pathogenic variants in *ENPP1* can cause classic PXE phenotype. In this study, two adult subjects, patients #11 and #12, from unrelated families (#6 and #7), had confirmed diagnoses of PXE and biallelic *ENPP1* sequence variants.

Family #6 had one affected subject of German descent, patient #11, a female aged 57 years (Fig 1F). Since 42 years of age, she has been a member of PXE International, the premier

**Table 1. Phenodex scores of patients diagnosed with PXE.**

| Patient | Age (years) | Phenodex scores | | | | |
|---------|-------------|-----------------|---|---|---|---|
| | | Skin (S) | Eye (E) | Vascular (V) | Cardiac (C) | Gastrointestinal (GI) |
| #11 | 11 | 1 | 0 | 0 | 0 | 0 |
| | 30 | 1 | 2 | 0 | 0 | 0 |
| | 57 | 1 | 2 | 0 | 0 | 0 |
| #12 | 8 | 3 | 1 | 0 | 0 | 0 |
| | 17 | 3 | 2 | 0 | 0 | 0 |
| | 27 | 3 | 2 | 0 | 0 | 0 |

organization that advocates for affected individuals and families with PXE. Around age 11, she noticed lesions and papules on her neck, wrist, elbow crease, and groin that have not significantly progressed in severity in the ensuing years (Fig 2F1). Histopathology of the lesional skin revealed calcification in the dermis, the characteristic feature of PXE. At the age of 30 years, she had peau d'orange and angioid streaks in both eyes (Fig 2F2), and at age 55, she had an ocular stroke and lost vision in her left eye. Cardiac and vascular examinations did not reveal anomalies. Blood work was normal (S1 Table). Only skin and eye had manifestations with Phenodex scores (Table 1; Detailed Phenodex scoring in each organ system [19]–S1: papules/bumps; S3: lax and redundant skin; E1: peau d'orange; E2: angioid streaks). The height of the patient, mother, and father was 5'1", 5', and 6', respectively. There was no evidence of bowed legs or rickets.

The proband in Family #7, a 27-year-old male, patient #12 of Caucasian and African American descent, was adopted at birth (Fig 1G). He has been a member of PXE International since he was 12. The adoptive parents are clinically healthy. Patient #12 noticed papules and bumps on the neck and around the umbilicus at age 6 years. His skin was lax and redundant with loss of recoil (Fig 2G1). A biopsy was taken from the lesional skin when he was age 8 years. A mineralization-specific stain, von Kossa, revealed calcium phosphate deposition in the mid dermis, characteristic of PXE. Peau d'orange was noticed at age 8 in his right eye. At age 10, he had angioid streaks in both eyes (Fig 2G2 and 2G3). Only skin and eye were affected with Phenodex scores (Table 1). His height is 6' and no evidence of bowed legs or rickets. Multiple blood analyses from 14 to 21 years of age showed normal mineral homeostasis (S1 Table).

## Identification of *ENPP1* variants in GACI and PXE patients

GACI patient #1 had a homozygous c.876_880delTAAAG (p.S292Rfs*4) variant in *ENPP1* (Fig 1A). This variant was not previously described. Parents are heterozygotes of this variant and clinically healthy. GACI patient #2 was homozygous for c.1756G>A (p.G586R) (Fig 1B), a previously reported variant in *ENPP1* [3,20]. Parents and an older son are heterozygotes. Due to family history of GACI, sequencing of fetal DNA demonstrated that patient #3 was homozygous for c.1756G>A, leading to the diagnosis of GACI prenatally (Fig 1B). Siblings #4, #5 and #6 with GACI had compound heterozygous, c.1441C>T (p.R481W) and c.2713_2717delAAAGA (p.K905Afs*16) variants, in *ENPP1* (Fig 1C). Both variants have been previously reported [3,20]. Parents and three other children are heterozygotes. Sibling GACI patients #7, #8, and cousin #9 were homozygous for a previously unreported variant, c.241G>T (p.V81L), in *ENPP1* (Fig 1D). Parents are heterozygotes, and twin sisters have wild-type alleles. GACI patient #10 had previously unreported compound heterozygous variants, c.656G>A (p.G219E) (paternal) and c.715+5G>T (maternal), in *ENPP1* as well as one heterozygous *ABCC6* variant, c.2477T>C (p.L826P), inherited from his unaffected mother (Fig 1E).

Biallelic variants in *ENPP1* were also identified in two patients diagnosed with classic PXE, patients #11 and #12. A previously unreported homozygous variant, c.1530G>C (p.L510F), was identified in patient #11 (Fig 1F1). One previously reported variant, c.1412A>G (p. Y471C) [20], and one previously unreported variant, c.2596G>A (p.E866K), were identified in patient #12 (Fig 1G).

## Uniparental disomy, upd(6)mat, in one family affected by PXE

Unexpectedly, variant segregation analysis in family #6 revealed Mendelian inconsistency for the c.1530G>C (p.L510F) variant in *ENPP1*. While patient #11 was homozygous for this variant, her mother was heterozygous, and her father was homozygous for the wild-type allele (Fig 1F1). Exome sequencing of the genomic DNA of patient #11 and her parents was performed for multiple purposes. First, we used exome sequencing to interrogate the presence of the c.1530G>C (p.L510F) variant in *ENPP1*, initially identified by a 29-gene ectopic calcification sequencing panel. Over 82,000 variants were annotated. Subsequent bioinformatic analyses included variant filtering for exonic and/or splice site variants, removal of synonymous and benign synonymous variants with CADD scores less than 20, removal of variants with minor allele frequency higher than 0.001, and removal of variants shared by the parents. Four candidate variants in each of the four genes, *RAPGEF2*, *ENPP1*, *SRCIN1*, and *MRPH8*, survived the stepwise bioinformatic filtering (Fig 1F2). The *ENPP1* gene is the only one known to cause ectopic calcification among the four candidate genes. The variant in *ENPP1*, c.1530G>C (p. L510F), was confirmed in patient #11. Secondly, we used exome sequencing for kinship analysis of patient #11 and her father. By algorithm, the estimated kinship coefficient of 0.177–0.354 indicates a first-degree relationship [21]. The kinship coefficient was calculated as 0.205 between patient #11 and her father, thus suggesting a first-degree relationship and excluding non-paternity. Finally, we used exome sequencing to investigate the possibility of uniparental disomy (UPD) in patient #11 – whether the c.1530G>C (p.L510F) variant on Chromosome 6 was inherited maternally (Fig 1F3). Maternal UPD on Chromosome 6 was identified based on variant alleles with these features: heterozygous in mother's DNA, homozygous in the patient's DNA, and homozygous wild-type allele in the father's DNA. Three Runs of Homozygosity (ROH) spanning 8.5 Mb, 14.6 Mb, and 11.0 Mb on Chromosome 6 were identified in patient #11's DNA, and the c.1530G>C (p.L510F) variant in *ENPP1* was located in the second ROH (Fig 1F3). Variants in these ROH segments were identical to those in the mother's DNA, and variants in regions outside of the three ROHs showed biparental inheritance. No ROH was found in the father's DNA. Thus, the inheritance pattern of mixed maternal UPD, upd(6)mat, was established in family #6.

## Bioinformatic analyses of *ENPP1* variants

A total of ten *ENPP1* variants were identified in 10 GACI patients and 2 PXE patients (Table 2). These variants are rare with minor allele frequency lower than 0.02% in the general population genetics databases–gnomAD and BRAVO. The aggregated predictions were deleterious for three variants only, c.656G>A (p.G219E), c.1756G>A (p.G586R), and c.715+5G>T. However, discrepancies were observed when various prediction tools were used. The American College of Medical Genetics and Genomics/Association for Molecular Pathology (ACMG/AMP) classifies the majority as variants of unknown significance (VUS) and three, c.876_880delTAAAG, c.1441C>T (p.R481W) and c.1756G>A (p.G586R), as likely pathogenic (LP). All variants had CADD scores above 20, predicted to be among the top 1% most deleterious to the human genome. These CADD scores were greater than the *ENPP1* gene-specific CADD score of 13.13 with 95% confidence interval [22], suggesting the intolerance of these variants.

**Table 2.** *In silico* predictions of *ENPP1* variants and comparison with experimental results. The five previously unreported variants in *ENPP1* were italic and underlined.

| Variant | Bioinformatic analyses | | | | Experimental results | | | | Pathogenicity |
|---|---|---|---|---|---|---|---|---|---|
| | Population data: No. of homozygous; MAF (%) | | Prediction outcome by | | ENPP1 expression (relative to the WT protein) | Subcellular localization | NPP activity (relative to the WT protein) | PPi in medium (relative to the WT protein) | |
| | gnomAD | BRAVO | Aggregated outcome | ACMG/ AMP | CADD score | | | | | |
| **Deletions (n = 2)** | | | | | | | | | | |
| *c.876_880delTAAAG, p. S292Rfs*4* | - | - | - | LP | 34.0 | Not expressed | - | Complete loss | Complete loss | Pathogenic |
| c.2713_2717delAAAGA, p.K905Afs*16 | 0; 0.0209 | 0; 0.0015 | - | VUS | 36.0 | - | - | Significantly reduced | - | Pathogenic |
| **Missense variants (n = 6)** | | | | | | | | | | |
| *c.656G>A, p.G219E* | - | - | Deleterious | VUS | 28.3 | Similar | IC | Complete loss | Complete loss | Pathogenic |
| c.1412A>G, p.Y471C | 0; 0.0085 | 0; 0.0087 | Uncertain | VUS | 24.7 | Similar | PM | Significantly reduced | Residual level | Pathogenic |
| c.1441C>T, p.R481W | 0; 0.0008 | - | Uncertain | LP | 26.3 | | | Significantly reduced | - | Pathogenic |
| | | | | | | Partial skipping of exon 15 (128 bp) | | | | |
| *c.1530G>C, p.L510F* | - | 0; 0.0004 | Uncertain | VUS | 22.8 | Similar | PM + IC | Significantly reduced | Significantly reduced | Pathogenic |
| c.1756G>A, p.G586R | 0; 0.0004 | 0; 0.0015 | Deleterious | LP | 27.3 | Significantly reduced | IC | Complete loss | Complete loss | Pathogenic |
| c.2596G>A, p.E866K | 0; 0.0035 | 0; 0.0045 | Uncertain | VUS | 26.9 | Similar | PM | Significantly reduced | Complete loss | Pathogenic |
| **Splicing variants (n = 2)** | | | | | | | | | | |
| *c.241G>T, p.V81L* | - | - | Uncertain | VUS | 24.4 | Complete skipping of exon 2 (73 bp) | | | | Pathogenic |
| *c.715+5G>T* | - | - | Deleterious | VUS | 25.7 | Complete skipping of exon 6 (98 bp); no protein synthesized | | | | Pathogenic |

## Functional characterization of five previously unreported *ENPP1* variants

In previous studies, five of the variants, c.1756G>A (p.G586R), c.1441C>T (p.R481W), c.2713_2717delAAAGA, c.1412A>G (p.Y471C) and c.2596G>A (p.E866K), were found to be pathogenic [7,20,23]. In addition, the c.1441C>T (p.R481W) variant was also found to cause partial skipping of exon 15 [7]. We analyzed the functional consequences of the five remaining, previously unreported variants, as described below.

RT-PCR followed by Sanger sequencing using patient #7's blood leukocytes showed that the c.241G>T variant (VUS) in exon 2 caused skipping of exon 2 (Fig 3A). The c.715+5G>T variant (VUS), identified in patient #10, was found to cause skipping of exon 6 in an *in vitro* mini-gene splicing assay (Fig 3B). Skipping of exon 2 or 6 is predicted to result in out-of-frame translation and generation of truncated and non-functional ENPP1 protein.

The functionality of the c.656G>A (p.G219E) (VUS), c.876_880delTAAAG (LP) and c.1530G>C (p.L510F) (VUS) variants was analyzed in transfected HEK293 and COS7 cells. The results showed that all three variants had deleterious effects on the ENPP1 protein. Specifically, the mutant proteins carrying p.G219E and p.L510F were expressed at similar levels to the wild-type (WT) protein, whereas no protein was detected for c.876_880delTAAAG (Fig 3C). The WT protein was localized predominantly on the plasma membrane (PM), the physiologic location of ENPP1 (Fig 3D). The p.G219E mutant showed intracellular localization (IC) while the p.L510F mutant showed both plasma membrane and intracellular localization (Fig 3D). The p.G219E and c.876_880delTAAAG mutants completely abolished enzyme activity, whereas p.L510F showed significantly reduced, about 60% of the WT protein (Fig 3E). Upon

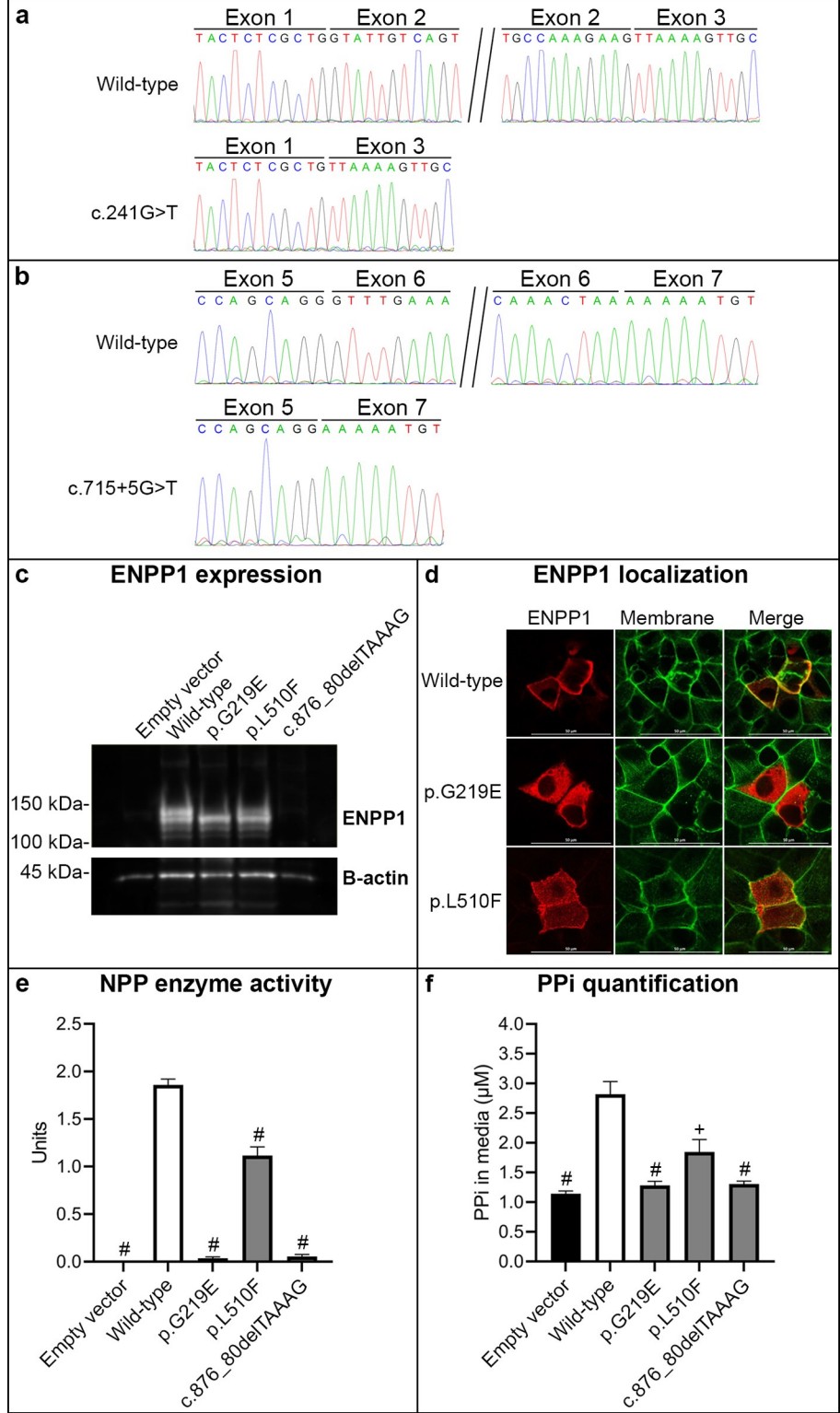

**Fig 3. Functional characterization of five previously unreported *ENPP1* variants, c.241G>T (p.V81L), c.715 +5G>T, c.656G>A (p.G219E), c.1530G>C (p.L510F) and c.876_880delTAAAG.** (**a**) Sanger sequencing of RT-PCR products from leukocytes in patient #7 carrying the c.241G>T variant showed skipping of exon 2. (**b**) The splicing of *ENPP1* mini-gene pre-mRNA from transfected HEK293 cells was assessed by RT-PCR using a forward primer specific to the mini-gene and a reverse primer targeting the vector's Flag-tag sequence, which allows splicing of the mini-gene-

produced transcripts to be studied. Sanger sequencing of RT-PCR products showed normal splicing for the wild-type construct and exon 6 skipping for mutant c.715+5G>T *ENPP1* mini-gene. (**c**) Western blot of ENPP1 protein expression in transfected HEK293 cells. ENPP1 monomers migrate as doublets at 118 kDa and 128 kDa on SDS-PAGE. Beta-actin was used as internal loading control. n = 3 independent experiments. (**d**) ENPP1 localization (red fluorescence) in transfected COS7 cells. Phalloidin, a peptide specific to actin filaments which are frequently found attached to or near the plasma membrane, was stained in green fluorescence to define plasma membrane. n = 3 independent experiments. Scale bar = 50 μm. (**e**) ENPP1 enzyme activity in transfected HEK293 cells, three cultures per group. $^{\#}P < 0.001$, compared with the wild-type construct. n = 3 independent experiments. $^{\#}P < 0.001$, compared with the wild-type construct. Data were presented as mean ± SEM. (**f**) Extracellular PPi generation in medium of transfected HEK293 cells 20 min after adding 20 μM GTP. Data represent % of PPi generated by the WT ENPP1-transfected cells. n = 3 independent experiments. $^{+}P < 0.01$, $^{\#}P < 0.001$, compared with the wild-type construct. Data were presented as mean ± SEM.

addition of 20 μM GTP in media of transfected cells, extracellular PPi in cells transfected with the WT construct increased significantly (Fig 3F). Extracellular PPi levels in cells expressing the p.G219E and c.876_880delTAAAG mutants were low, similar to the empty vector (Fig 3F). The p.L510F mutant was still able to generate PPi, but its amount was significantly lower than the WT protein (Fig 3F).

Collectively, all *ENPP1* variants were functionally assessed to be pathogenic (Table 2).

## Circulating concentrations of PPi in GACI and PXE patients carrying biallelic *ENPP1* variants

Plasma PPi concentrations were determined in patients #3, 6, 11, and 12, all harboring biallelic pathogenic variants in *ENPP1*. The results demonstrated that these individuals, regardless of the diagnosis of GACI or PXE, showed significantly reduced PPi plasma concentrations (Fig 4). Heterozygotes had intermediate levels of PPi. A family member, the father of patient #11, carrying homozygous wild-type alleles, had PPi concentration similar to healthy controls (Fig 4).

## Discussion

GACI, regardless of ABCC6- or ENPP1-deficiency, is a life-threatening arterial calcification disease with a poor prognosis. Although there are several reports of long-term survivors into their third to fifth decade [3,6,24], a large proportion of children with GACI die within the first six months of life. Death is related to cardiovascular collapse, including myocardial infarction, congestive heart failure, persistent pulmonary hypertension, and multi-organ failure. GACI survivors with ENPP1-deficiency develop ARHR2 with short stature and skeletal deformities [3,18]. It was suggested that ARHR2 is FGF23-mediated [3], and the elevated serum FGF23 levels in several GACI patients in the current study support this hypothesis. Elevated serum FGF23 levels may also be a response of cells to circulating calciprotein particles, which are associated with vascular calcification [25]. ABCC6-associated PXE has a different clinical course with late-onset and more favorable clinical outcomes than GACI. Though fully penetrant, clinical features of PXE, including skin, eye, and vasculature lesions, do not usually present until adolescence or early adulthood. In contrast to GACI, individuals affected by PXE have normal life expectancy in the overwhelming number of cases, without evidence of skeletal anomalies.

Studies over the past few years have indicated that ENPP1- and ABCC6-deficiency are associated with considerable clinical pleiotropy. Specifically, some patients with *ENPP1* variants develop hypophosphatemic rickets without a prior clinical history of GACI [26,27], while nearly all patients with *ABCC6* variants present with PXE without a history of GACI [2]. In the current study, while all the patients harbor biallelic *ENPP1* variants, their varied clinical

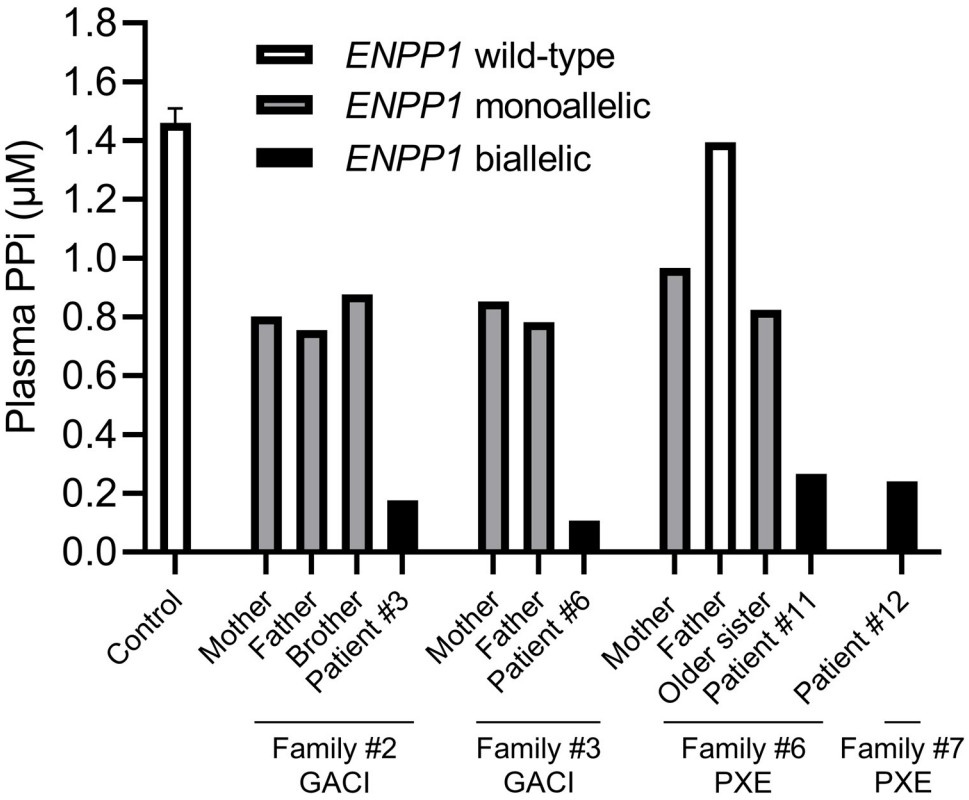

**Fig 4. Plasma PPi concentrations in individuals with GACI or PXE and family members.** Individuals with biallelic *ENPP1* pathogenic variants, regardless of the diagnosis of GACI or PXE, had barely detectable PPi plasma concentrations. *ENPP1* heterozygotes had intermediate PPi plasma concentrations. The physiologic PPi plasma concentrations were obtained from 9 healthy control volunteers. Data represent mean ± SEM.

expression was highlighted by them seeking support from different advocacy organizations. Among the 10 GACI patients enrolled in GACI Global, eight presented with extensive calcifications of large and medium-sized arteries in the prenatal or neonatal period, while two GACI patients, #7 and #10, had arterial stenosis without evidence of vascular calcification. This is not surprising as regression of vascular calcification has been documented in patients with GACI [24,28]. By contrast, two adult patients #11 and #12 had classical PXE, and both were enrolled in the PXE International registry. From a clinical point of view, these patients do not differ from the typical PXE patients with ABCC6-deficiency. Our studies demonstrate that in addition to GACI and rickets, ENPP1-deficiency can also present as classical PXE, a finding that extends the clinical spectrum of ENPP1-associated diseases. Moreover, because the two *ENPP1* variants identified in PXE patient #12 had previously been described in patients diagnosed as GACI [20,23], there does not appear to be a genotype-phenotype explanation for the variation in these two clinical presentations.

In this study, we identified five previously unreported *ENPP1* variants. We also report original findings of UPD inheritance of a previously unreported homozygous variant, c.1530G>C (p.L510F). The c.241G>T and c.715+5G>T variants were functionally determined to cause aberrant splicing, although c.241G>T (p.V81L) was initially thought to be a missense variant. In contrast to splicing variants, different *ENPP1* coding variants have different outcomes on the protein's functionality. These include reduced protein abundance, impaired cellular localization, compromised stability and/or conformational changes, reduced enzyme activity, and

ability to generate PPi. Although the c.1530G>C (p.L510F) mutant had reduced enzyme activity and PPi generation, its residual activity was probably attributed to the partial localization on the plasma membrane. Therefore, the p.L510F variant appears to be a candidate for allele-specific therapy in correcting the misfolded, otherwise functional protein, as previously demonstrated for ABCC6 [29–31].

While plasma PPi concentrations in patients with ABCC6-deficiency were reduced to approximately 30–50% of controls, patients with ENPP1-deficiency had a further reduction to approximately 0–10% of controls. It is not clear why the same *ENPP1* variants can result in different diagnoses of either PXE or GACI since our patients' plasma PPi concentrations were equally low. Several potential mechanisms may explain the phenotypic heterogeneity and the poor correlation between plasma PPi concentrations and disease severity. First, environmental factors and genetic modifiers may influence the disease severity of ectopic calcification [2,32]. Second, circulating PPi may be a poor proxy of the local PPi concentrations which may be more important in preventing tissue calcification. Unfortunately, we cannot currently measure extracellular PPi levels in tissues. Third, in addition to PPi, ENPP1-mediated hydrolysis of ATP also produces adenosine monophosphate. The pathophysiologic role of adenosine monophosphate in the disease process of GACI was recently reported [9]. Furthermore, the potential dysregulation of extracellular nucleotide metabolism, for example, ENPP1-mediated disruption of pyrimidine synthesis known to regulate tissue repair, may play a role in ectopic tissue calcification [33]. The concept of ENPP1-deficiency has evolved dramatically over the past several decades: what once considered an exclusively fatal arterial disease with poor prognosis is now recognized as a complex, multi-systemic process with a broad phenotypic spectrum spanning from infantile vascular calcification associated with early demise, to hypophosphatemic rickets in survivors, and as indicated in this study, to typical late-onset PXE with more favorable prognosis and normal life expectancy.

In conclusion, the phenotypic spectrum of ENPP1-deficiency is much broader than was previously anticipated. In addition to GACI, we show that the late-onset skin and ocular phenotypes of patients with ENPP1-deficiency can be indistinguishable from typical PXE with ABCC6-deficiency. The divergent phenotypes in patients with ENPP1-deficiency cannot be explained exclusively by plasma concentrations of PPi which were reduced to the same extent. The results suggest that although PPi is a major determinant of ectopic calcification, additional yet unidentified mechanisms may play a role in the regulation of ectopic calcification.

## Materials and methods

### Ethics statement

All patients were enrolled with written or verbal informed consent/assent into this study with approval from the institutional review board at Children's Hospital of Philadelphia (Approval number 12–008863) or Genetic Alliance (Approval number PXE001 for PXE International). Formal consent was obtained from the parent/guardian for child participants. Patients with GACI and PXE were registered in the databases of GACI Global and PXE International, advocacy organizations for GACI and PXE, respectively. We used the Phenodex score, an international standard to assess phenotypes in five organ systems: skin (S), eyes (E), gastrointestinal (G), cardiac (C), and vasculature (V), to determine the clinical severity of PXE [19].

### Variant detection and bioinformatics

Genomic DNA was extracted from saliva (DNA Genotek Inc) or peripheral blood leukocytes (Qiagen, Valencia, CA). Variant detection was performed by Sanger sequencing of the entire coding region and intron/exon boundaries of the *ABCC6* and *ENPP1* genes, exome sequencing

(MyGenostics, Beijing, China), next-generation sequencing of hypophosphatemic rickets-targeted genes including *ENPP1* (Exeter Clinical Laboratory, Leeds, UK), or next-generation sequencing of ectopic calcification-associated 29 gene panel including *ABCC6* and *ENPP1* [34]. Exome sequencing and stepwise bioinformatics were performed according to previously reported approaches [35,36]. The kinship analysis was done using VCFtools—relatedness2 on the merged Variant Call Format files [21]. The screening for Runs of Homozygosity (ROH) of more than 4 Mb and establishment of patterns of UPD in trio samples were performed according to the previously described method, with slight modifications [37].

*ENPP1* variant nomenclature was based on NC_000006.12 (NM_006208). The variant nomenclature followed the recommendations of the Human Genome Variation Society (http://www.hgvs.org/mutnomen/). The number of individuals carrying the specific variant as homozygous and the minor allele frequency in the general population was extracted from Genome Aggregation Database (gnomAD) (gnomad.broadinstitute.org) and BRAVO (https://bravo.sph.umich.edu/freeze8/hg38/) consisting of over 120,000 and 130,000 apparently healthy individuals, respectively. The recommended *ENPP1* gene-specific MAF threshold is 0.1% (https://franklin.genoox.com/clinical-db/home). Various *in silico* prediction programs (https://franklin.genoox.com/clinical-db/home) and the Combined Annotation Dependent Depletion (CADD) score were used to assess the effects of variants on the protein function [38,39]. Classification of variants follows the latest guidelines of the American College of Medical Genetics and Genomics/Association for Molecular Pathology (ACMG/AMP), which classifies variants as benign (B), likely benign (LB), variants of unknown significance (VUS), likely pathogenic (LP), and pathogenic (P) [40,41]. The *ENPP1* gene-specific CADD score within the 95% confidence interval was calculated using the mutation significance cutoff method [22].

## RNA analysis of the c.241G>T (p.V81L) variant in *ENPP1*

Total RNA was extracted from patient #7's peripheral blood cells after venous blood collection in a PAXgene Blood RNA tube (BD Diagnostics, Franklin Lakes, NJ) followed by RT-PCR and Sanger sequencing.

## *In vitro* mini-gene splicing assay of the c.715+5G>T variant in *ENPP1*

WT and c.715+5G>T mutant mini-gene segments spanning from exon 5 to exon 7 of human *ENPP1* were cloned into the pCMV-3Tag-3a vector (Genscript, Piscataway, NJ). Human embryonic kidney (HEK293) cells were transfected with WT or mutant constructs using FuGENE HD (Promega, Madison, WI). Cells were collected 48 hours post-transfection for RNA extraction and RT-PCR followed by bidirectional Sanger sequencing of different transcript isoforms.

## Functional assessment of c.656G>A (p.G219E), c.1530G>C (p.L510F), and c.876_880delTAAAG variants in *ENPP1*

The full-length WT cDNA and mutant human *ENPP1* cDNA entailing each of the three variants, c.656G>A, c.1530G>C, and c.876_880delTAAAG, were cloned into a pcDNA3.1(+) vector (GenScript). HEK293 and *Cercopithecus aethiops* kidney (COS7) cells were transfected using jetPEI (Illkirch, France). We measured the activity of nucleotide phosphodiesterase (NPP) 24 hours after transfection of HEK293 cells using pNP-TMP as substrate [7,20]. Expression of human ENPP1 protein was detected by Western blot using a rabbit anti-human ENPP1 antibody, #5342, 1:1,000 (Cell Signaling, Boston, MA). An anti-human β-actin antibody, #4970, 1:1,000, was used to normalize protein loading (Cell Signaling). The cellular localization of the ENPP1 protein was analyzed in transfected COS7 cells using a monoclonal anti-

human ENPP1 antibody, 1:100 (3E8, kind gift from Dr. Fabio Malavasi, Torino, Italy). Cells were stained with fluorescein isothiocyanate labeled phalloidin, #P5282, 1:40 (Sigma-Aldrich, Taufkirchen, Germany), to visualize plasma membrane localization. The PPi concentration in the medium of transfected HEK293 cells was measured 20 min after incubation with 20 μM GTP. PPi was quantified as previously described [10,12].

## Biochemical analyses

The serum concentrations of calcium, phosphorus, alkaline phosphatase, fibroblast growth factor 23, parathyroid hormone, and 25-hydroxyvitamin D3 were retrieved from patients' medical records. Whole blood was collected into CTAD and transferred to EDTA tubes (BD Diagnostics), followed by depletion of platelets by filtration through a Centrisart I 300-kDa mass cutoff filter (Sartorius, New York, NY). Determination of PPi concentration in platelet-free plasma was performed as previously described [10,12].

## Statistical analysis

Statistical analyses were performed using ordinary one-way ANOVA. Statistical significance was considered with $P < 0.05$. All statistical analyses were completed using Prism 8 (Graph-Pad, San Diego, CA).

## Supporting information

**S1 Text. Clinical features and biochemical findings of GACI patients.**
(DOCX)

**S1 Table. Biochemical findings of patients with *ENPP1* variants.**
(DOCX)

## Acknowledgments

We thank all the affected individuals and families for their participation. We thank Dr. Hansel J. Otero at Children's Hospital of Philadelphia, and Dr. Meisam Sargazi in Alzahra Eye Hospital Research Center at Zahedan University of Medical Sciences, Zahedan, Iran, for interpretations of clinical images. We thank Mary Peckiconis at PXE International for assistance in obtaining the participants' medical records, and Dr. Talat Mushtaq at Leeds Children's Hospital in UK for assistance with the clinical findings in family #4.

## Author Contributions

**Conceptualization:** Qiaoli Li.

**Data curation:** Douglas Ralph, Yvonne Nitschke, Michael A. Levine, Matthew Caffet, Tamara Wurst, Amir Hossein Saeidian, Leila Youssefian, Sharon F. Terry, Frank Rutsch, Jouni Uitto, Qiaoli Li.

**Formal analysis:** Douglas Ralph, Yvonne Nitschke, Michael A. Levine, Matthew Caffet, Tamara Wurst, Amir Hossein Saeidian, Leila Youssefian, Hassan Vahidnezhad, Sharon F. Terry, Frank Rutsch, Jouni Uitto, Qiaoli Li.

**Funding acquisition:** Jouni Uitto, Qiaoli Li.

**Investigation:** Douglas Ralph, Yvonne Nitschke, Michael A. Levine, Matthew Caffet, Tamara Wurst, Amir Hossein Saeidian, Leila Youssefian, Hassan Vahidnezhad, Sharon F. Terry, Frank Rutsch, Jouni Uitto, Qiaoli Li.

**Methodology:** Qiaoli Li.

**Project administration:** Qiaoli Li.

**Resources:** Michael A. Levine, Sharon F. Terry, Jouni Uitto, Qiaoli Li.

**Software:** Douglas Ralph, Amir Hossein Saeidian, Leila Youssefian, Qiaoli Li.

**Supervision:** Qiaoli Li.

**Validation:** Qiaoli Li.

**Visualization:** Qiaoli Li.

**Writing – original draft:** Qiaoli Li.

**Writing – review & editing:** Douglas Ralph, Yvonne Nitschke, Michael A. Levine, Matthew Caffet, Tamara Wurst, Amir Hossein Saeidian, Leila Youssefian, Hassan Vahidnezhad, Sharon F. Terry, Frank Rutsch, Jouni Uitto, Qiaoli Li.

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
