## [Decision Letter · Decision Letter 0]

24 Feb 2022

Dear Dr Li,

Thank you very much for submitting your Research Article entitled 'ENPP1 variants in patients with GACI and PXE expand the clinical and genetic heterogeneity of heritable disorders of ectopic calcification' to PLOS Genetics.

The manuscript was fully evaluated at the editorial level and by independent peer reviewers. The reviewers appreciated the attention to an important topic but identified some concerns that we ask you address in a revised manuscript

We therefore ask you to modify the manuscript according to the review recommendations. Your revisions should address the specific points made by each reviewer.

[LINK]

Yours sincerely,

Melissa Wasserstein, MD

Associate Editor

PLOS Genetics

Gregory Barsh

Editor-in-Chief

PLOS Genetics

Reviewer's Responses to Questions

**Comments to the Authors:**

Reviewer #1: The authors reports the clinical, laboratory, and molecular evaluations of ten GACI and two PXE patients from five and two unrelated families registered in GACI Global and PXE International databases, respectively. The authors conclude that the phenotypic spectrum of ENPP1-deficiency is much broader than was previously anticipated. It is known that GACI and PXE are complex disease, and the genes involved have many modifiers, e.g. doi: 10.3389/fcell.2021.612581. The authors further state that the correlation of plasma PPi and severity of GACI and PXE may not hold. The content of the paper could be greatly increased if the authors discuss alternative disease mechanisms beside the long-held association of extracellular PPi and calcification inhibition. Circulating PPi may be a poor proxy of the local PPi concentrations, which may actually determine the cellular calcification milieu. Also, cleavage of each ATP releases AMP along with PPi. What ist known about the role of AMP signalling in these diseases? Adenine, and specific ribonucleosides that disrupt pyrimidine synthesis may regulate the severity of GACI and PXE by affecting cell survival. (Li et al https://doi.org/10.1172/JCI149711). Please discuss.

li 151 "The Family #7 had one adopted 27-year-old male, patient #12, of Caucasian and African American descent (Fig. 1g). He also seeks support from PXE International." The fact that an unrelated adopted child acquired similar affection to me suggests the contribution of environmental or nutritional factors. Is this possible or was the child adopted BECAUSE it was affected? Please explain.

li 272 "elevated serum FGF23 levels in several GACI patients in the current study support this hypothesis" FGF-23 is a phosphatonin. Elevated levels may suggest phosphate and calciprotein particle toxicity with consequences for cell ageing and cell death (https://doi.org/10.1016/j.kint.2019.10.019 and papers cited therein). Please discuss.

Reviewer #2: Authors of the work entitled “ENPPI variants in patients with GACI and PXE expand the clinical and genetic heterogeneity of heritable disorders of ectopic calcification” present a thorough and interesting article describing how mutations in ENPP1, classically described in cases of GACI, are also present in patients with PXE. While GACI is seen as a severe form of soft tissue calcification disorder, PXE is considered “less severe” given its later onset in life and relatively reduced impact on patient morbidity and mortality. This work demonstrates that mutations in ENPP1 can also lead to phenotypes indicative of “less severe” PXE, illustrating clear genetic heterogeneity and resulting phenotypes across patients.

A clear highlight of this work is how the authors utilize two patient networks to obtain data and support detailed genetic and molecular analysis. This work would not have been possible without the support of these networks and illustrates an important role they play in the scientific community.

Minor edits to the work are as follows:

1. For readers less familiar with this pathway, please provide a graphical representation for the pathways of interest in the introduction highlighting ABCC6, ENPP1, Pyrophosphate, ATP, etc.

2. In the introduction both GACI1 and GACI are utilized- please keep consistent

3. On line 170 it indicates that patients 7, 8, and 9 are siblings. From the diagram in figure 1, is patient 9 a sibling or cousin? Please correct.

4. An additional paragraph in the results section connection PPi levels to phenotype is warranted. This is a main idea of the abstract and could be better discussed in the results, along side the PPi level measures. Are Phenodex type of values available for GACI patients?

5. Please change blue arrows in Figure 2 to another color- maybe yellow, to increase visibility

6. Please provide scale bars throughout Figure 2

7. At the end of figure 2 legend, it denotes d, dead; a, alive – where is this denoted in the figure? All patients included alive, correct?

8. Please provide scale bars for 3d.

9. Please indicate how many biological or technical replicated were performed in Figure 3.

10. Please expand in the methods the concentration of antibody used for both western and fluorescent cell-based analysis. 1:100? 1:1000? Additional details to help other reproduce such work is needed.

**Have all data underlying the figures and results presented in the manuscript been provided?**

Reviewer #1: Yes

Reviewer #2: Yes

PLOS authors have the option to publish the peer review history of their article (what does this mean?). If published, this will include your full peer review and any attached files.

Reviewer #1: **Yes: **Willi Jahnen-Dechent

Reviewer #2: No

---

## [Editor Report · Decision Letter 1]

5 Apr 2022

Dear Dr Li,

We are pleased to inform you that your manuscript entitled "ENPP1 variants in patients with GACI and PXE expand the clinical and genetic heterogeneity of heritable disorders of ectopic calcification" has been editorially accepted for publication in PLOS Genetics. Congratulations!

Yours sincerely,

Melissa Wasserstein, MD

Associate Editor

PLOS Genetics

Gregory Barsh

Editor-in-Chief

PLOS Genetics

Comments from the reviewers (if applicable):

**Data Deposition**

http://datadryad.org/submit?journalID=pgenetics&manu=PGENETICS-D-21-01563R1

**Press Queries**

---

## [Editor Report · Acceptance letter]

25 Apr 2022

PGENETICS-D-21-01563R1 

*ENPP1* variants in patients with GACI and PXE expand the clinical and genetic heterogeneity of heritable disorders of ectopic calcification 

Dear Dr Li, 

We are pleased to inform you that your manuscript entitled "*ENPP1* variants in patients with GACI and PXE expand the clinical and genetic heterogeneity of heritable disorders of ectopic calcification" has been formally accepted for publication in PLOS Genetics! Your manuscript is now with our production department and you will be notified of the publication date in due course.

With kind regards,

Agnes Pap

PLOS Genetics

On behalf of:
